Anthropogenic reverberations on the gut microbiome of dwarf chameleons (Bradypodion)

Adair Matthew G. mattgadair@gmail.com 1 2
Tolley Krystal A. 1 2
van Vuuren Bettine Jansen 2
da Silva Jessica Marie 1 2
1 Kirstenbosch Research Centre, South African National Biodiversity Institute , Cape Town , Newlands , South Africa
2 Centre for Ecological Genomics and Wildlife Conservation, University of Johannesburg , Johannesburg , Gauteng , South Africa
Uversky Vladimir
Electronic publication date: 2025 Feb 28
Publication date: 2025
Volume: 13
Electronic Location ID: e18811
Received 2024 Jul 5; Accepted 2024 Dec 12
Copyright: ©2025 Adair et al.
Copyright year: 2025
Copyright holder: Adair et al.
License: This is an open access article distributed under the terms of the Creative Commons Attribution License, which permits unrestricted use, distribution, reproduction and adaptation in any medium and for any purpose provided that it is properly attributed. For attribution, the original author(s), title, publication source (PeerJ) and either DOI or URL of the article must be cited.
License URL: https://creativecommons.org/licenses/by/4.0/

Keywords: Metabarcoding, Small reptiles, Bacteriome, Microbiota, Natural vs. urban habitats, South Africa

Funding: National Research Foundation (NRF) of South Africa, Dimensions of Biodiversity Grant Program 136381 This research was funded by National Research Foundation (NRF) of South Africa, Dimensions of Biodiversity Grant Program (grant no. 136381). The funders had no role in study design, data collection and analysis, decision to publish, or preparation of the manuscript.

==============================
Exploration of the microbiome has been referred to as a final frontier in biological research. This is due to its precedence for generating insights on the holistic functioning of organismal biology by exploring the interactions between hosts and their associated symbiotic organisms. The microbiomes of many vertebrate groups still require exploration to advance current knowledge and fill previous knowledge gaps. This study generated initial descriptions of the bacterial microbiomes of three species of dwarf chameleon (Bradypodion) from the 16S rRNA gene region targeting the V3 and V4 hypervariable regions. This led to the successful identification of 1,073 and 4,502 independent amplicon sequence variants from buccal swab and faecal material samples, respectively. This newly acquired information is intended as a baseline for future work incorporating holobiont information. The diversity of microbial taxa suggests that the total dwarf chameleon microbiome is similar to other squamates investigated to date, as well as chelonians (Testudines). Microbial frequency differences were noted in comparison to crocodilians (Archosauria) and mammalian groups. Furthermore, this study aimed to examine the influence of habitat transformation on the composition of the microbiome in dwarf chameleons as each of the study species occupy both urban and natural habitats. Given that most urban habitats are highly transformed, the expectation was that microbial assemblages of the gastro-intestinal tracts of all three Bradypodion species would show significant differences between populations (i.e., natural, or urban). It was found, however, that the level of effect was contingent on species: B. melanocephalum populations showed noticeable microbiome differences between urban and natural populations; B. thamnobates showed variations in microbial community dispersions between populations; and B. setaroi showed no significant microbiome differences based on diversity metrics although some frequency differences, in microbiome composition, were observed between populations. We suggest that the magnitude of difference between the habitats occupied by the populations is a factor, given the apparent disparity between the natural and urban habitats for B. melanocephalum as compared to the other two species.

Introduction

The natural world houses a prodigious display of co-diversification between symbiotic microbes and their hosts; both plants (Bouffaud et al., 2014; Abdelfattah et al., 2022) and animals (Ley et al., 2008; Rosenberg & Zilber-Rosenberg, 2016) show a unique symbiotic specificity with their microbial communities. These symbiotic relationships can have a Daedalian complexity, where fractional alteration of the microbial communities can be the difference between healthy and diseased individuals (Schumacher, 2006; Trevelline et al., 2019). Environmental factors can play an important part in influencing the composition of a microbiome (Baca et al., 2012; Murtaza et al., 2019) to such an extent that the environment can be even more important than host genotype in dictating which microbes form a symbiotic relationship with the host (Hyde et al., 2016; Rothschild et al., 2018; Zhou et al., 2020; San Juan, Castro & Dhami, 2021; Fieschi-Méric et al., 2023).

The fundamental importance of the environment to the microbiome stems from the fact that the proximate environment is a major inoculation source for host microbial communities (Holgerson et al., 2013), the other being vertical transmission from mother to foetus or neonate (e.g., Baca et al., 2012). Furthermore, the greater the spatiotemporal heterogeneity within the environment, the greater the biodiversity within the environment, even at a microbial level (Chesson, 2000; Oliver et al., 2010; Shade et al., 2012; Yang et al., 2015; Walters & Martiny, 2020). This relationship is especially prevalent when comparing the microbiomes of natural and captive individuals, as the reduced environmental heterogeneity of captivity tends to lead to poorer microbiome health (Hyde et al., 2016; San Juan, Castro & Dhami, 2021; Fieschi-Méric et al., 2023) or can even alter microbial taxa frequencies (Zhou et al., 2020).

Owing to the fact that anthropogenic activities tend to disrupt and replace stable natural conditions with a homogenised ecological structure (Newbold et al., 2018), they pose one of the most serious threats facing ecosystems today (Pimm et al., 2014; Ceballos et al., 2015; Newbold et al., 2016). Current forecasts of future environmental conflict paint a sobering picture; anthropogenic activity is expected to intensify in the near future (Laurance et al., 2014) and, as a consequence, a greater number of species are being confronted with anthropogenic habitat transformation. This confrontation leads to a dilemma; either species can adapt to the neoteric environment, or they can leave the altered environment completely through immigration or extinction (Burger & Lynch, 1995; Winchell et al., 2020).

The defining characteristics that separate which species are likely to persist and which are likely to go extinct as a consequence of anthropogenic activities remain elusive due to the wide range of possibilities and contingencies (Johnson & Munshi-South, 2017). This, however, is not always the case as some species show a penchant for adaptation to anthropogenic modification (Johnson & Munshi-South, 2017; Winchell et al., 2020). Furthermore, contemporary hypotheses, such as the Hologenome Theory of Evolution (Zilber-Rosenberg & Rosenberg, 2008; Rosenberg & Zilber-Rosenberg, 2016), indicate that the simple explanation of singular-organismal adaptation to changing environments may not be the only solution found in nature. Following the Hologenome hypothesis, one would expect to uncover significant differences in microbiome assemblage between organisms which inhabit urban environments and those that still reside in natural ones.

Insights into the complex and distinct host-microbial assemblages of reptile microbiomes is now burgeoning (Jiang et al., 2017; Weitzman, Gibb & Christian, 2018; Ross, Rodrigues Hoffmann & Neufeld, 2019; Arizza et al., 2019; Zhou et al., 2020; Siddiqui, Maciver & Khan, 2022; Bunker, Arnold & Weiss, 2022; Vasconcelos et al., 2023) and many of their finer aspects remain to be explored (see Colston, 2017). The benefits of a more holistic examination of reptilian microbiomes would include the potential to unveil novel information regarding host-microbe interaction (i.e., Moon & Lee, 2016; Davoodi & Foley, 2020; Emami-Khoyi et al., 2020). Evidence already indicates that the microbiomes of many species are influenced by being in captivity (Alfano et al., 2015; Zhou et al., 2020) or by exposure to anthropogenic activity (Amato et al., 2013). These shifts in microbiota could translate into dysbiosis in the animals (Schumacher, 2006), which could influence the effects of successful reintroductions. Furthermore, the microbiome can have fundamental physiological effects on the host, even changing the ability of the host to tolerate temperature changes (Su, Zhou & Zhang, 2013; Moeller et al., 2020). The application of this information in conservation could unveil a previously unconsidered aspect during any animal handling, which would be to consider the far-reaching effects of alterations to the microbiome assemblage (Redford et al., 2012; Bahrndorff et al., 2016).

To date, the microbiomes of chameleons have not been characterised and an examination of the role of environmental influence on the microbiota of chameleons still remains obscure. Typically, chameleons are insectivorous and mainly employ ambush foraging as a hunting strategy (Keren-Rotem, Bouskila & Geffen, 2006; Measey et al., 2011; Carne & Measey, 2013; da Silva et al., 2016; Stanton-Jones et al., 2024). It has been observed that prey choice and habitat occupation in some animals strongly correlates with their contained microscopic communities (Kohl et al., 2017; Dion-Phénix et al., 2021). Further evidence supports a strong correlation between the skin microbiome of captive reptiles to their restrained environment (Hyde et al., 2016), with a general homogenisation of the microbiome assemblage (Passos, Garcia & Young, 2018). Therefore, due to differences in distributions and habitat preferences (and hence, assumed corresponding differences in available prey), it can be predicted that differences may be present in the microbial diversity between natural and urban populations of dwarf chameleons.

The three species examined here each inhabit different vegetation types within KwaZulu-Natal (KZN) Province, South Africa (Fig. 1), with distributions that include differing degrees of habitat modification. Bradypodion melanocephalum (Gray, 1865) occupies a variety of habitats from grassy savanna to riparian thickets, along the edges of aquatic systems (wetlands, lotic systems), as well as in heavily transformed urban landscapes that include green-belts, private gardens, and road verges (Tolley & Burger, 2007; Tilbury, 2018). Bradypodion thamnobates (Raw, 1976) occupies closed canopy indigenous forest patches (Tolley & Burger, 2007; Tilbury, 2018), as well as moderately transformed areas such as green-belts, private gardens, and road verges within a larger matrix of an agricultural landscape. Bradypodion setaroi (Raw, 1976) occurs in coastal dune forests in the south-eastern coast of KZN stretching into the southern-most areas of Mozambique but also occurs in minimally transformed areas where remnants of indigenous forest vegetation persist within small towns (Tolley & Burger, 2007; Tilbury, 2018).

Figure 1 Species distribution map with sampled areas.

Respective species distributions (solid polygons) and sampling areas (scored polygons within each distribution) for Bradypodion melanocephalum sampled in and around the city of eThekwini (refer to Fig. S1 for detailed view of sample area); B. setaroi sampled in and around the town of St Lucia (refer to Fig. S2 for detailed view of sample area); and B. thamnobates (refer to Fig. S3 for detailed view of sample area) sampled in and around the town of Howick. Overlaid on the respective biomes within KwaZulu-Natal, South Africa: grassland, forests, savanna, Indian ocean coastal belt, and azonal vegetation.

Perhaps as a consequence of the diversity of habitats with which Bradypodion can be associated, the group often displays clear ecomorphological variation within some species (Hopkins & Tolley, 2011; da Silva & Tolley, 2013; da Silva & Tolley, 2017; Tolley, Hopkins & da Silva, 2019). Some species even show a strong presence in urban landscapes (Tolley, Hopkins & da Silva, 2019; Petford et al., 2024) that represent conceivably novel populations. The independence of these populations is still unclear; however, radio tracking has shown limited displacement of Bradypodion pumilum individuals (Rebelo et al., 2022), which could suggest limited immigration among distant populations. Furthermore, as the microbiome holds close ties to habitat occupation of the host organism, dwarf chameleons offer particular interest as a group due to their high adaptability to different environments. Combined with this, Bradypodion as a genus is facing a daunting rate of anthropogenic change (Tolley et al., 2019) making the group ideally stationed for generating knowledge about anthropogenic habitat change to microbiome constitution.

To examine the effects of anthropogenic change on the microbiome assemblage of dwarf chameleons, initial microbiomes of B. melanocephalum, B. setaroi, and B. thamnobates were characterised. Bacterial phylogenetic trees (representative of the collective microbiomes of each species) and abundance plots were constructed from the massive parallel sequencing of the 16S rRNA V3 and V4 regions of both buccal and faecal samples (gamma diversity). Using buccal swab and faecal samples the overall bacterial diversity in the gastro-intestinal tract was compared (alpha diversity) between species. It was expected that each species would show a unique microbiome assemblage but that many taxa would be conserved between species. The influence of habitat occupation on the microbiome was explored through the comparison of urban and natural populations (beta diversity) within each species. The expectation was that microbial assemblages would show stark detectable differences between natural or urban populations.

Methodology and materials

Ethics approval

All animal handling and sample collection was approved by the University of Johannesburg (ethics no.: 2019-10-10/van Vuuren_Tolley) and provincial permits from KwaZulu-Natal (OP2635/2020). Portions of this text were previously published as part of a thesis (Adair, 2023).

Sample collection

Buccal and faecal sampling of B. melanocephalum, B. thamnobates, and B. setaroi individuals was carried out in situ (Fig. 1). Dwarf chameleon individuals were located at night with the use of torchlight. Once an individual was located and retrieved locality information (WGS 84, at ∼11 m precision) was recorded using a GPS receiver unit (Garmin MAP65), and a piece of marked chevon tape was placed on the same tree or bush to allow for release at the exact site of capture. For oral microbial sampling, 20 buccal swabs were collected per species (10 from natural and 10 from urban habitats) upon locating and retrieving an individual (Figs. S1–S3). Sterile gloves were worn for the swabbing of each individual with gloves being changed between individuals. Sterile cotton FLOQswabs (Zymo DNA/RNA Shield Collection Tube w/ Swab) were used for the collection of saliva with the complete buccal cavity being swabbed, including the outer gums, to ensure full coverage. This was achieved by coercing the chameleon to open its mouth and bite onto the swab before rotating the swab for approximately 1 min. Each buccal swab was then immediately immersed in the tube containing DNA/RNA shield reagent supplied by the manufacturers, the extended swab handle broken off at the moulded breaking point, and the tube sealed and labelled. All samples were transported to the research base at ambient temperature. Once at the research base, the sealed collection tubes were stored at −4 °C.

Upon completion of swabbing and the collection of the correlated metadata, each chameleon was transported in cloth bags containing foliage from the capture site, until they and their foliage could be transferred to a 3.3 L clear container at the research base. Each individual was held for a 24-hour period to allow for the collection of faecal samples prior to release at the exact site of collection. Assuming that not all swabbed chameleons would defecate before being released, additional chameleons were collected to ensure an adequate sample size. As such, not all faecal samples correspond exactly to individuals sampled with buccal swabs. A total of 60 faecal samples—with 20 samples per species (10 from natural and 10 from urban habitats)—were collected. Faecal collection was achieved with a pair of stainless-steel forceps which were wiped down, bleached (10% v/v), and flamed with 95% ethanol, between collections. Each individual’s faecal matter was placed into separate tubes containing nucleic acid preservation buffer and stored at −4 °C. Both buccal swab and faecal samples were transported on ice from the research station to the storage facility where they were stored at −40 °C prior to further processing.

Laboratory processing

Total genomic DNA was extracted from the 60 buccal swabs using a Zymo Quick-DNA Fecal/Soil MiniPrep Kit following the protocol provided in the user manual with minimal alteration. Deviation from the protocol included: allowing the swabs to sit in the lysis buffer at 55 °C for 10 mins prior to DNA extraction, as well as subsequently adding 200 µl of swab storage buffer to increase total genomic DNA yield. The total genomic DNA of the 60 faecal samples was extracted with the use of a Zymo Quick-DNA Fecal/Soil Microbe MiniPrep Kit following the protocol provided in the user manual. Approximately 150 mg of faecal material was used for each extraction. All DNA was eluted in a final volume of 50 µl with the provided elution buffer.

The hypervariable regions V3 and V4 of the 16S rRNA gene region were amplified using F341 (5′-CCT ACG GGA GGC AGC AG-3′) (Muyzer, de Waal & Uitierlinden, 1993) and R806 (5′-GGA CTA CHV GGG TWT CTA AT-3′) (Walters et al., 2011) primers, respectively. All amplification PCRs were completed in 20 µl reactions consisting of: 10 µl of Thermo Fisher Scientific Platinum Multiplex PCR Master Mix, 4 µl of 2 µM pooled forward and reverse primers, up to 5 µl of Milli-Q H2O, and 25–50 ng/µl of gDNA template. PCR amplification consisted of initial denaturation at 94 °C for 3 mins, followed by 25 cycles of 94 °C for 30 s, 55 °C for 30 s, and 72 °C for 30 s, followed by a final extension at 72 °C for 5 mins.

All amplicons were cleaned using AMPure XP beads in a ratio of 0.8 AMPure XP beads to amplicon volume following a slightly adjusted cleaning protocol, including: an initial dilution of 10 µl of sample to 40 µl ddH2O. Cleaned amplicons were eluted in a final volume of 25 µl of 10 mM Tris at pH 8.5. Proceeding amplicon clean-up, all samples were indexed in 50 µl reactions consisting of: 25 µl of 2 × KAPA HiFi HotStart ReadyMix, five µl of Nextera XT Index 1 Primers (N7XX), five µl of Nextera XT Index 2 Primers (S5XX), 10 µl of Milli-Q H2O, and five µl cleaned PCR amplicons. PCR amplification consisted of initial denaturation at 95 °C for 3 mins, followed by eight cycles of 95 °C for 30 s, 55 °C for 30 s, and 72 °C for 30 s, followed by final extension at 72 °C for 5 mins. Indexed products were then cleaned using AMPure XP beads in a ratio of 1.12 AMPure XP beads to amplicon volume following the cleaning protocol. Subsequent equimolar pooling of samples was managed by the South African Institute for Aquatic Biodiversity (SAIAB) prior to sequencing.

Sequencing

All amplified samples were sequenced at the South African Institute for Aquatic Biodiversity (SAIAB) with the use of an Illumina MiSeq machine in a 2 × 300 bp run. The initial data was demultiplexed at the sequencing facility and stored in FASTQ formatted files. All data files were uploaded to the Ilifu high performance computing facility hosted by the University of Cape Town’s Information and Communications Technology Services (http://www.ilifu.ac.za) for the subsequent analyses. New DNA sequences generated for this study were deposited on GenBank (BioProject accession number PRJNA1184952).

Quality control

Initial quality control checks of all raw sequence reads were made with the use of FastQC v0.11.9 (Andrews, 2010) and a compiled visualisation of the quality controls was made using MultiQC (Ewels et al., 2016). This output was used to denoise all raw-sequence data in QIIME2 v2022.2.1 (Bolyen et al., 2019). Truncation of primers occurred to 250 bp with the first 20 bp of each sequence being trimmed to remove primer binding site contamination. Subsequent denoising (in the form of quality filtering, pair merging, and chimeric removal) of the raw sequence features was achieved under the DADA2 (Callahan et al., 2016) plugin in QIIME2. This resulted in the retention of high-quality amplicon sequence variants (ASVs), which were then aligned using MAFFT (Katoh et al., 2002).

Analysis of primary microbiome diversity

Samples were analysed in groups either belonging to buccal swabs or faecal pellets due to the different natures of the treatments. Sequence identification across all ASVs was made using the inbuilt q2-feature-classifier (Bokulich et al., 2018) used in tandem with the reference sequences of the pretrained 515f-806r-animal-distal-gut-classifier.qza classifier (Quast et al., 2013; Kaehler et al., 2019; Robeson et al., 2021), which was generated off the Silva v138 database in preference to animal digestive tracts. Sequence identification informed further data filtering to ensure the removal of miscellaneous sequences including any matches to “mitochondria”, “chloroplasts”, or “Archaea”. Furthermore, sequences which could not be identified to phylum level or lower taxonomic rank were discarded. Following this, phylogenetic trees were constructed with the use of FastTree2 (Price, Dehal & Arkin, 2010) plugin based on sample type—either buccal swab or faecal material. Subsequent visualisation of phylogenetic trees occurred using the EMPress (Cantrell et al., 2021) plugin in QIIME2, allowing for visual comparison of bacterial microbiome diversity between sample types. Visual representations of the frequency of microbial phyla were generated in the form of heatmaps using the Matplotlib (Hunter, 2007) plugin.

Alpha diversity metrics were calculated to describe the bacterial diversity within each sample type per species. Faith’s phylogenetic diversity described the observed bacterial diversity accounting for respective branch lengths in the phylogenetic tree (Faith, 1992). Pielou’s evenness described whether certain bacterial taxa are dominating the observed diversity (Pielou, 1966). Shannon’s diversity described the level of bacterial diversity within each species (Shannon & Weaver, 1949). Kruskal-Wallis pairwise statistics (Kruskal & Wallis, 1952) allowed for the comparison of alpha diversity metrics between Bradypodion species with q-values being calculated to estimate the likelihood of positive false discovery rate in multiple hypothesis testing. To identify which microbial taxa were differentially abundant between Bradypodion species ANalysis of COmpositions of Microbiomes (ANCOM) (Mandal et al., 2015) were run. No threshold value was set as the ANCOM program uses an approach that empirically derives the threshold for significant differences based on the supplied dataset. The ANCOM tests were focused at each taxonomic level to indicate at what rank microbial taxa are most differentially abundant. Furthermore, a search through the identified ASVs for Aeromonas, Campylobacter, Edwardsiella, Escherichia, Flavobacterium, Klebsiella, Mycobacterium, Salmonella, and Serratia was made to determine if any of these known zoonotic bacterial taxa were present in the dataset (Schumacher, 2006; Institutional Animal Care and Use Committee, 2021).

Analysis of habitat effects on microbiome composition

Comparisons between urban and natural populations were first made by sorting independent sequences by species and then categorising them by population. To identify which microbial families were differentially abundant between habitat types ANCOMs were run followed by ANCOMs with Bias Correction (ANCOM-BC) (Lin & Peddada, 2020) using natural populations as the reference for comparison to urban populations. The significance threshold of the ANCOM-BC testing was set to 0.05.

Following this, standard beta diversity metrics were then computed in QIIME2 using a sampling depth of 1,000 based on the alpha rarefaction curves. The beta diversity metrics used included, Bray-Curtis dissimilarity (Sørensen, 1948) to compare the difference in diversity between two systems, Jaccard similarity index (Jaccard, 1908) to compare the overlap and distinctness in species diversity between two systems, unweighted UniFrac distance (Lozupone & Knight, 2005) and weighted normalised UniFrac distance (Lozupone et al., 2007) both of which compare differences in diversity based on phylogenetic information, with the former considering presence/absence data and the latter considering relative frequency data. Visualisations of all beta diversity metrics occurred as Principal Coordinates Analysis (PCoA) with the use of EMPeror (Vázquez-Baeza et al., 2013).

Beta diversity metrics were statistically compared between urban and natural populations of each species using PERmutational Multivariate analysis of DISpersion (PERMDISP) (Anderson, 2001) to distinguish whether dispersion was driving community differences. Single-factor PERmutational Multivariate ANalysis Of VAriance (PERMANOVA) were also calculated with p-values corrected as q-values to accommodate for multiple hypothesis testing in the PERMANOVA. All PERMDISP and PERMANOVA tests were carried out for 999 permutations following the 3 dimensions depicted in the standard PCoA plots. All other parameters were retained at default. Multifactorial ANOVAs were run in the form of Adonis testing (Anderson, 2001; Oksanen et al., 2018) to establish the significance of the role of host-species and habitat in the composition of the microbiome. Primarily, habitat was run with species as an interaction parameter but to ensure rigidity of results species was run with habitat as an interaction item.

Results

Primary microbiome diversity

Sequencing resulted in 10,358,081 raw sequence reads across all 120 chameleon (buccal and faecal) samples. Subsequent denoising resulted in 898,960 raw sequence reads being retained, which were compiled into 1,073 and 4,502 unique sequence features (ASVs) for buccal swab and faecal pellet samples, respectively. Subsequent identifications using the trained classifier assigned taxonomic ranks to all ASV identifications and resulted in unique identifications of 25 phyla, 38 classes, 87 orders, 158 families, 276 genera, and 347 species. Removal of miscellaneous sequences through filtering of non-bacterial identifications resulted in the retainment of 1,065 (99.25%) (Table S1) and 4,458 (99.02%) (Table S2) ASVs from buccal swab and faecal material samples, respectively.

Heatmap visualisations of bacterial phyla frequency for buccal swabs (Fig. 2) vs faecal material (Fig. 3) showed clear dominance of bacterial phyla across all Bradypodion. Examination of the relative frequency of ASVs identified to phylum level showed differences in the composition of the most frequent phyla relative to sample type. Buccal swab samples for all chameleon species were dominated by Proteobacteria (x ¯ ± SD = 51.22 ± 7.84%), with Firmicutes (x ¯ ± SD = 35.08 ± 6.53%) following closely in terms of relative frequency (Figs. 2 & 4). Bacteroidota (x ¯ ± SD = 12.01 ± 6.50%) (synonymous with Bacteroidetes) was the third most frequent phylum in the buccal swab samples for all three species. Actinobacteriota (x ¯ ± SD = 1.17 ± 0.99%), and Fusobacteriota (x ¯ ± SD = 0.36 ± 0.12%) were the next most abundant phyla across all buccal swab samples. Faecal samples were all dominated by Firmicutes (x ¯ ± SD = 42.27 ± 3.09%) (Figs. 3 & 5). Proteobacteria (x ¯ ± SD = 26.93 ± 12.05%) and Bacteroidota (x ¯ ± SD = 20.35 ± 5.96%) were the next most frequent phyla, with Proteobacteria being the second most abundant phylum in B. melanocephalum and B. thamnobates, whilst Bacteroidota was the second most frequent phylum in B. setaroi. Verrucomicrobiota (x ¯ ± SD = 4.03 ± 3.51%), Fusobacteria (x ¯ ± SD = 2.92 ± 2.20%), Spirochaetota (x ¯ ± SD = 1.69 ± 1.22%), Actinobacteriota (x ¯ ± SD = 0.93 ± 0.23%), and Desulfobacterota (x ¯ ± SD = 0.76 ± 0.56%) were the next most abundant phyla across all faecal samples.

Figure 2 Heatmap representation of the log(10) relative frequencies of the identified bacterial phyla across buccal swab samples.

Relative frequency of each identified bacterial phylum is represented along a colour gradient corresponding to the logarithmic transformation of the initial frequency value. Independent samples are displayed by column with Bradypodion species indicated by bracketed sections.

Figure 3 Heatmap representation of the log(10) relative frequencies of the identified bacterial phyla across faecal material samples.

Relative frequency of each identified bacterial phylum is represented along a colour gradient corresponding to the logarithmic transformation of the initial frequency value. Independent samples are displayed by column with Bradypodion species indicated by bracketed sections.

Figure 4 Phylogenetic tree showing the retained bacterial ASV composition of all buccal swab samples for each Bradypodion species, with relative ASV frequency plots.

Relative level of abundance of each ASV identification between Bradypodion species is indicated in the surrounding purple-blue band, correlated to indicated species colour. Further independent taxa bar plots showing relative phyla abundance for each respective sample are indicated.

Figure 5 Phylogenetic tree showing the retained bacterial ASV composition of all faecal pellet samples for each Bradypodion species, with relative ASV frequency plots.

Relative level of abundance of each ASV identification between Bradypodion species is indicated in the surrounding purple-blue band, correlated to indicated species colour. Further independent taxa bar plots showing relative phyla abundance for each respective sample are indicated.

Alpha diversity of buccal swab samples revealed no statistical differentiation in Shannon indices, as well as Faith’s phylogenetic diversity across all three Bradypodion species (Table 1). Mean Shannon index and mean Faith’s phylogenetic diversity was highest in B. melanocephalum followed by B. setaroi, with B. thamnobates showing the lowest values (Fig. 6). Pielou’s evenness was high across all three species (x ¯ > 0.75) indicating that the bacterial community is mostly homogeneous with a few taxa having slightly higher frequencies in the buccal cavity. Faecal samples showed a high similarity in Shannon indices (Fig. 7), and Faith’s phylogenetic diversity between B. melanocephalum and B. setaroi, however both these species had statistically significant differences compared to B. thamnobates (Table 1). Mean Shannon index, and Faith’s phylogenetic diversity was highest in B. thamnobates followed by B. melanocephalum and then B. setaroi. Pielou’s evenness in faecal samples was similar to that found in the buccal cavity across all three species (i.e., x ¯ > 0.75), indicating that the bacterial community is mostly homogeneous with a few taxa having slightly higher frequencies in the hindgut.

Differentially abundant taxa were examined at all taxonomic ranks to assess which groups may be driving diversity differences between dwarf chameleon species. ANCOM testing found no significant differences among taxa in the buccal swab samples (Fig. 8) suggesting conserved microbial communities in buccal cavities of the three Bradypodion species. In contrast, several differentially abundant taxa were found in the faecal samples (Fig. 9). The greatest number of differentially abundant taxa were identified at family level with 10 independent families showing high differential abundance. Five phyla were identified as differentially abundant at the phylum level, four classes at class level, three orders at order level, six genera at genus level, and one species at species level.

Lastly, the exploration of zoonotic bacterial genera within the identified ASVs found no hits for six of the nine targeted genera (Aeromonas, Edwardsiella, Flavobacterium, Klebsiella, Mycobacterium, and Salmonella) across all ASV (both buccal and faecal) identifications. Three of the searched bacterial genera (Campylobacter, Escherichia, and Serratia) had a small portion of representatives identified in the faecal ASVs; however, no matches were found in the identified buccal ASVs.

Habitat effects on microbiome composition

The relationship between habitat occupation and microbiome composition was explored in the three Bradypodion species to assess whether occupation of transformed habitats has a correlation to the bacterial microbiome composition in the dwarf chameleon digestive tract. Individuals were sampled from both urban and natural populations (Figs. S1–S3) allowing for specific habitat distinctions to be made between sample groups. Buccal and faecal samples were analysed separately in light of the differences observed during the initial microbiome description.

Table 1 Kruskal–Wallis (H test) results for alpha diversity comparisons between Bradypodion species.

Faith’s phylogenetic diversity, Pielou’s evenness, and Shannon diversity alpha diversity metrics were tested. Presented are the species being compared, H, p-values, and q-values for each parameter. Significant comparisons are indicated in bold.

Metric	Sample type	Species 1	Species 2	H	p-value	q-value	
Faith’s phylogenetic diversity	Swab	B. melanocephalum	B. setaroi	0.208	0.649	0.649	
B. melanocephalum	B. thamnobates	1.032	0.310	0.464	
B. setaroi	B. thamnobates	4.233	0.040	0.119	
Faecal	B. melanocephalum	B. setaroi	0.178	0.673	0.673	
B. melanocephalum	B. thamnobates	20.898	<0.001	<0.001	
B. setaroi	B. thamnobates	15.253	<0.001	<0.001	
Pielou’s evennes	Swab	B. melanocephalum	B. setaroi	3.188	0.074	0.184	
B. melanocephalum	B. thamnobates	0.088	0.767	0.767	
B. setaroi	B. thamnobates	2.381	0.123	0.184	
Faecal	B. melanocephalum	B. setaroi	0.256	0.613	0.613	
B. melanocephalum	B. thamnobates	6.059	0.014	0.042	
B. setaroi	B. thamnobates	2.938	0.087	0.130	
Shannon diversity	Swab	B. melanocephalum	B. setaroi	1.214	0.271	0.406	
B. melanocephalum	B. thamnobates	0.088	0.767	0.767	
B. setaroi	B. thamnobates	2.542	0.111	0.332	
Faecal	B. melanocephalum	B. setaroi	0.133	0.715	0.715	
B. melanocephalum	B. thamnobates	10.891	0.001	0.003	
B. setaroi	B. thamnobates	5.704	0.017	0.025	

Figure 6 Alpha diversity distribution plots for buccal swab samples.

Distribution plots for the alpha diversity metrics calculated across buccal swab samples: (A) Shannon’s diversity index; (B) Faith’s phylogenetic diversity index; (C) Pielou’s evenness index. Displayed per species: Bradypodion melanocephalum, B. setaroi, and B. thamnobates.

Figure 7 Alpha diversity distribution plots for faecal material samples.

Distribution plots for the alpha diversity metrics calculated across buccal swab samples: (A) Shannon’s diversity index; (B) Faith’s phylogenetic diversity index; (C) Pielou’s evenness index. Displayed per species: Bradypodion melanocephalum, B. setaroi, and B. thamnobates.

Figure 8 Volcano plots for ANCOM results at each taxonomic level for all buccal swab samples.

Differential abundance shown at bacterial taxonomic rank (A) phylum; (B) class; (C) order; (D) family; (E) genus; (F) species.

Figure 9 Volcano plots for ANCOM results at each taxonomic level for all faecal pellet samples.

Differential abundance shown at bacterial taxonomic rank (A) phylum; (B) class; (C) order; (D) family; (E) genus; (F) species. Significantly different taxon names are displayed.

Initial bacterial phylum frequencies showed varying levels of difference between urban and natural populations for the respective sample types (Table 2). Bradypodion melanocephalum had the lowest variations in bacterial phyla frequencies between natural and urban populations (Firmicutes x ¯ variation ± SD = 5.70 ± 5.85%; Proteobacteria x ¯ variation ± SD = 15.01 ± 3.03%; Bacteriodota x ¯ variation ± SD = 2.86 ±0.86%). Both B. setaroi (Firmicutes x ¯ variation ± SD = 17.74 ± 15.09%; Proteobacteria x ¯ variation ± SD = 10.19 ± 6.57%; Bacteriodota x ¯ variation ± SD = 7.17 ± 7.07%) and B. thamnobates (Firmicutes x ¯ variation ± SD = 21.23 ± 22.22%; Proteobacteria x ¯ variation ±SD = 20.20 ± 11.51%; Bacteriodota x ¯ variation  ± SD = 4.82 ± 3.37%) showed significantly more variation in bacterial phyla frequencies.

Differential abundance testing (ANCOM) was carried out at family level to compare differences between habitats (Figs. 8 & 9), as initial ANCOMs of all buccal swabs and faecal material samples indicated family as the most differentially abundant taxonomic level. Few bacterial families were identified as differentially abundant between populations and sample types (Fig. 10). Notably Enterococcaceae was identified as differentially abundant in buccal cavities between populations of B. melanocephalum, whilst Ruminococcaceae was differentially abundant in the hindgut between populations of B. thamnobates. To ensure biases were not driving the lack of differences, ANCOM-BCs were run. These also showed minimal differences between populations; however, a few notable families were identified as differentially abundant. Caulobacteraceae was shown to be enriched, whilst Enterococcaceae was depleted in the buccal cavities of urban B. melanocephalum. Desulfovibrionaceae was enriched in the hindguts of urban B. melanocephalum. Desulfovibrionaceae, Christensenellaceae, Ruminococcaceae, and Akkermansiaceae were all enriched in the hindguts of urban B. thamnobates.

Principal coordinate analyses of the four calculated beta diversity metrics showed minimal clustering of habitat samples upon visual examination (Figs. S4–S6). PERMDISP comparisons of calculated beta diversity metrics found no major differences in sample dispersion between the sampled populations of either B. melanocephalum or B. setaroi for both buccal swab and faecal material samples (Table 3). Significant differences (p ≤ 0.05) were noted in the Bray-Curtis dissimilarity, Jaccard similarity, and unweighted UniFrac distance comparison in the faecal samples of B. thamnobates; however, all beta diversity comparisons of buccal swab samples as well as weighted UniFrac distance of faecal samples in B. thamnobates had no statistical significance. The PERMANOVA calculations comparing the beta diversity of urban and natural populations of B. melanocephalum found significant differences (q ≤ 0.05) across all four beta diversity metrics calculated for faecal samples; however, only the unweighted UniFrac distance was significant for the beta diversity metrics from buccal swabs of B. melanocephalum (Table 4). Bradypodion setaroi had no significance for any of the metrics except for the Bray-Curtis dissimilarity calculated for faecal samples. Significant differences were also noted for the Jaccard similarity and unweighted UniFrac distance in the faecal samples from B. thamnobates; however, all other metrics for B. thamnobates had no significant differences.

Adonis tests run on all four beta diversity metrics using habitat as the test variable and species as an interaction parameter consistently attributed microbiome differences in the samples to host-species of origin rather than habitat differences between populations within each species (Table 5). To ensure parameter ordering was not a confounding factor in the adonis calculations, tests were run with species as the test variable, while habitat was treated as an interaction item (Table 6). These produced the same outcome indicating that host-species is a main driver of microbiome differences across the samples whilst habitat has a lesser influence on microbiome composition.

Discussion

Primary microbiome diversity

The first descriptions of bacterial microbiome communities found within the digestive tracts (foregut and hindgut) of dwarf chameleons were explored and visualised. The characterisation of these microbiomes forms the stepping stones for future research into the microbial communities found within chameleons, and specifically the genus Bradypodion, as this provides a baseline for comparison. More specifically, the described microbiomes could be used as a baseline for understanding dysbiosis within the genus. For the assignment of sequence identity to retained sequence features an ASV approach was implemented. This allowed for high-resolution during sequence identification (Caruso et al., 2019), whilst ensuring datasets are comparable both within and outside the current study (Callahan, McMurdie & Holmes, 2017). The buccal cavity microbiomes between the three examined species were remarkably homogeneous in overall bacterial community composition with no differentially abundant taxa being identified at any taxonomic rank by standard ANCOM testing. This may suggest a functional overlap in the bacterial communities found within the buccal cavities that could be limiting change. Tests of the faecal samples, however, indicate differential abundance at all taxonomic ranks.

Taxonomic assignment of identified ASVs showed diversity of faecal microbiota to be much higher than oral microbiota. This differs from studies of other vertebrate species (e.g., Kropáčková et al., 2017; Maki et al., 2021) and may suggest that chameleon-bacterial symbiosis has distinctive patterns of frequency compared to other vertebrates. Alpha diversity calculations showed consistently high levels of bacterial diversity for all three Bradypodion species that is comparable to other squamate lizards such as Anolis (Ren et al., 2016). Both anoles and chameleons show a notable degree of ecomorphological variation within species (Poe & Anderson, 2019). This shared trait may partially explain the similarities in microbiome diversity between the two groups. However, it must be noted that Anolis is highly species rich in comparison to Bradypodion, as well as much more thoroughly studied. Therefore, detailed future comparisons will have to be drawn to assess the degree of overlap between the microbiomes of these two genera. Comparisons to other reptilian groups suggest that dwarf chameleons, in terms of broad microbiome assemblage, appear to be typical within squamates (Hong et al., 2011; Ren et al., 2016; Kohl et al., 2017). Furthermore, noticeable similarities can be seen with dwarf chameleon microbiomes and those of chelonians (Testudines) (Yuan et al., 2015), and even birds (Aves) (Sun et al., 2022). There is divergence, however, between dwarf chameleons and crocodilians [Archosauria], which have digestive tracts that are typically dominated by Fusobacteria making them an exception rather than the rule in Reptilia (Keenan, Engel & Elsey, 2013).

Table 2 Relative frequencies (%) of identified bacterial phylum in urban and natural populations.

Shown are faecal material and buccal swab samples per Bradypodion species sampled. Frequencies the seven most frequent bacterial phyla are given. All other bacterial phyla are listed under ‘Other’.

Phylum	Relative frequency (%)	
	B. melanocephalum	B. setaroi	B. thamnobates	
	Faecal	Swab	Faecal	Swab	Faecal	Swab	
	Natural	Urban	Natural	Urban	Natural	Urban	Natural	Urban	Natural	Urban	Natural	Urban	
Firmicutes	39.77	41.33	11.15	20.98	42.03	49.10	50.54	22.13	34.89	40.41	54.81	17.88	
Proteobacteria	33.48	16.33	74.05	61.17	10.65	16.20	36.13	50.96	50.88	38.82	42.20	70.54	
Bacteroidota	17.12	19.37	13.33	16.79	29.66	27.49	12.37	24.54	13.54	11.10	2.38	9.59	
Fusobacteriota	6.81	5.81	0.35	0.64	2.83	1.68	0.00	0.49	0.34	4.60	0.29	1.06	
Actinobacteriota	0.82	1.07	0.42	0.33	0.88	1.33	0.96	1.87	0.36	1.01	0.18	0.76	
Desulfobacterota	0.63	2.37	0.49	0.01	1.36	0.64	N/A	N/A	<0.01	0.42	0.04	0.11	
Verrucomicrobiota	1.17	9.91	0.16	0.07	12.32	2.92	<0.01	<0.01	<0.01	0.77	0.03	0.03	
Other	0.21	3.81	0.07	0.00	0.28	0.64	<0.01	<0.01	<0.01	2.88	0.07	0.05	

Figure 10 Volcano plots for ANCOM results at Family taxonomic level for all samples.

Differential abundance shown at bacterial taxonomic rank (A) B. melanocephalum swab; (B) B. melanocephalum faecal; (C) B. setaroi swab; (D) B. setaroi faecal; (E) B. thamnobates swab; (F) B. thamnobates faecal. Significantly different taxon names are displayed.

Table 3 Results from PERMDISP comparing urban and natural populations of each Bradypodion species.

Significance values (p-values) for each beta diversity metric (Bray–Curtis dissimilarity index; Jaccard similarity index; Unweighted UniFrac distance; and Weighted UniFrac distance) are given for the buccal swab and faecal matter samples. Significant comparisons are indicated in bold.

Metric	Sample type	Sample size	F-value	p-value	
B. melanocephalum	
Bray–Curtis dissimilarity	Swab	18	0.097	0.743	
Faecal	20	0.773	0.261	
Jaccard similarity	Swab	18	1.272	0.274	
Faecal	20	0.115	0.718	
Unweighted UniFrac distance	Swab	18	1.039	0.310	
Faecal	20	1.901	0.179	
Weighted UniFrac distance	Swab	18	0.151	0.699	
Faecal	20	0.470	0.487	
B. setaroi	
Bray–Curtis dissimilarity	Swab	14	0.210	0.606	
Faecal	19	0.335	0.501	
Jaccard similarity	Swab	14	0.170	0.689	
Faecal	19	1.544	0.205	
Unweighted UniFrac distance	Swab	14	0.130	0.729	
Faecal	19	1.905	0.183	
Weighted UniFrac distance	Swab	14	0.480	0.376	
Faecal	19	0.128	0.683	
B. thamnobates	
Bray–Curtis dissimilarity	Swab	12	0.013	0.941	
Faecal	20	7.867	0.003	
Jaccard similarity	Swab	12	0.011	0.922	
Faecal	20	24.465	0.001	
Unweighted UniFrac distance	Swab	12	0.026	0.856	
Faecal	20	9.821	0.002	
Weighted UniFrac distance	Swab	12	0.012	0.894	
Faecal	20	2.395	0.131	

Table 4 Results from PERMANOVA comparing urban and natural populations of each Bradypodion species.

Significance values (p- and q-values) for each beta diversity metric (Bray–Curtis dissimilarity index; Jaccard similarity index; Unweighted UniFrac distance; and Weighted UniFrac distance) are given for the buccal swab and faecal matter samples. Significant comparisons are indicated in bold.

Metric	Sample type	Sample size	pseudo-F	p-value	q-value	
B. melanocephalum	
Bray–Curtis dissimilarity	Swab	18	0.861	0.585	0.585	
Faecal	20	2.085	0.001	0.001	
Jaccard similarity	Swab	18	1.124	0.209	0.209	
Faecal	20	1.452	0.001	0.001	
Unweighted UniFrac distance	Swab	18	1.928	0.020	0.020	
Faecal	20	2.631	0.001	0.001	
Weighted UniFrac distance	Swab	18	1.283	0.304	0.304	
Faecal	20	3.730	0.007	0.007	
B. setaroi	
Bray–Curtis dissimilarity	Swab	14	1.245	0.238	0.238	
Faecal	19	1.491	0.049	0.049	
Jaccard similarity	Swab	14	1.434	0.114	0.114	
Faecal	19	1.121	0.098	0.098	
Unweighted UniFrac distance	Swab	14	1.963	0.057	0.057	
Faecal	19	1.421	0.063	0.063	
Weighted UniFrac distance	Swab	14	1.222	0.297	0.297	
Faecal	19	0.981	0.422	0.422	
B. thamnobates	
Bray–Curtis dissimilarity	Swab	12	0.941	0.451	0.451	
Faecal	20	1.234	0.164	0.164	
Jaccard similarity	Swab	12	0.967	0.456	0.456	
Faecal	20	1.357	0.005	0.005	
Unweighted UniFrac distance	Swab	12	1.282	0.135	0.135	
Faecal	20	2.234	0.010	0.010	
Weighted UniFrac distance	Swab	12	1.352	0.225	0.225	
Faecal	20	1.098	0.330	0.330	

Table 5 Results from Adonis testing the significance of habitat differences when interacting with species (*Species) in Bradypodion microbiome composition from all samples.

Bray–Curtis dissimilarity index, Jaccard similarity index, Unweighted UniFrac distance, and Weighted UniFrac distance beta diversity were tested. Presented are the sums of squares, mean squares, F-value, r2, and p-values for each parameter. Significant comparisons are indicated in bold.

Metric	Parameter	Df	SumsOfSqs	MeanSqs	F-value	R 2	p-value	
Swab	
Bray Curtis dissimilarity	Habitat	1	0.372	0.372	1.128	0.023	0.312	
Species	2	2.340	1.170	3.551	0.146	0.001	
Habitat:Species	2	0.771	0.386	1.170	0.048	0.232	
Jaccard similarity	Habitat	1	0.387	0.387	1.125	0.024	0.265	
Species	2	1.813	0.906	2.633	0.112	0.001	
Habitat:Species	2	0.919	0.460	1.335	0.057	0.057	
Unweighted UniFrac distance	Habitat	1	0.313	0.313	1.539	0.032	0.088	
Species	2	0.979	0.490	2.410	0.101	0.001	
Habitat:Species	2	0.647	0.323	1.591	0.067	0.021	
Weighted UniFrac distance	Habitat	1	0.148	0.148	1.147	0.024	0.304	
Species	2	0.707	0.354	2.738	0.115	0.016	
Habitat:Species	2	0.368	0.184	1.425	0.060	0.199	
Faecal	
Bray Curtis dissimilarity	Habitat	1	0.770	0.770	1.928	0.031	0.001	
Species	2	1.856	0.928	2.324	0.074	0.001	
Habitat:Species	2	1.168	0.584	1.462	0.047	0.007	
Jaccard similarity	Habitat	1	0.479	0.479	2.461	0.035	0.007	
Species	2	2.156	1.078	5.535	0.158	0.001	
Habitat:Species	2	0.687	0.343	1.763	0.050	0.011	
Unweighted UniFrac distance	Habitat	1	0.479	0.479	2.461	0.035	0.007	
Species	2	2.156	1.078	5.535	0.158	0.001	
Habitat:Species	2	0.687	0.343	1.763	0.050	0.011	
Weighted UniFrac distance	Habitat	1	0.172	0.172	1.768	0.023	0.154	
Species	2	1.675	0.837	8.624	0.227	0.001	
Habitat:Species	2	0.392	0.196	2.019	0.053	0.065	

Table 6 Results from Adonis testing the significance of species differences when interacting with habitat (*Habitat) in Bradypodion microbiome composition from all samples.

Bray-Curtis dissimilarity index, Jaccard similarity index, Unweighted UniFrac distance, and Weighted UniFrac distance beta diversity were tested. Presented are the sums of squares, mean squares, F-value, r2, and p-values for each parameter. Significant comparisons are indicated in bold.

Metric	Parameter	Df	SumsOfSqs	MeanSqs	F-value	R 2	p-value	
Swab	
Bray Curtis dissimilarity	Species	2	2.382	1.191	3.615	0.149	0.001	
Habitat	1	0.330	0.330	1.001	0.021	0.411	
Species:Habitat	2	0.771	0.386	1.170	0.048	0.25	
Jaccard similarity	Species	2	1.832	0.916	2.662	0.113	0.001	
Habitat	1	0.367	0.367	1.068	0.023	0.298	
Species:Habitat	2	0.919	0.460	1.335	0.057	0.055	
Unweighted UniFrac distance	Species	2	0.990	0.495	2.436	0.102	0.001	
Habitat	1	0.302	0.302	1.486	0.031	0.098	
Species:Habitat	2	0.647	0.323	1.591	0.067	0.029	
Weighted UniFrac distance	Species	2	0.722	0.361	2.794	0.118	0.015	
Habitat	1	0.134	0.134	1.035	0.022	0.378	
Species:Habitat	2	0.368	0.184	1.425	0.060	0.205	
Faecal	
Bray Curtis dissimilarity	Species	2	1.871	0.936	2.343	0.075	0.001	
Habitat	1	0.754	0.754	1.889	0.030	0.004	
Species:Habitat	2	1.168	0.584	1.462	0.047	0.008	
Jaccard similarity	Species	2	1.635	0.818	1.915	0.063	0.001	
Habitat	1	0.649	0.649	1.519	0.025	0.001	
Species:Habitat	2	1.088	0.544	1.274	0.042	0.003	
Unweighted UniFrac distance	Species	2	2.165	1.083	5.557	0.159	0.001	
Habitat	1	0.471	0.471	2.416	0.034	0.013	
Species:Habitat	2	0.687	0.343	1.763	0.050	0.015	
Weighted UniFrac distance	Species	2	1.688	0.844	8.689	0.228	0.001	
Habitat	1	0.159	0.159	1.638	0.022	0.164	
Species:Habitat	2	0.392	0.196	2.019	0.053	0.053	

The buccal cavities of all three dwarf chameleon species were dominated by Proteobacteria, with Firmicutes being a close second. Proteobacteria (synonymous with Psuedomonadota) are a highly diverse phylum found throughout numerous environments and containing a plethora of pathogenic species (Schoch et al., 2020). In carnivorous mammals the phylum is associated with metabolism of proteins, amino acids, carbohydrates, and vitamins (Moon et al., 2018). This group could be playing similar functions in Bradypodion digestion making them indispensable symbiotes. Furthermore, because these bacteria tend to perform key functions in the digestive process, their high frequency is perhaps unsurprising. Crocodile lizards (Shinisaurus; Squamata) also typically have a high frequency of Proteobacteria (Jiang et al., 2017) making a comparable symbiotic relationship to dwarf chameleons.

The relative frequency of bacterial phyla within the faecal material of dwarf chameleons showed a functional shift in the community composition, as compared to the buccal cavity. Firmicutes become the most frequent bacterial phyla for all Bradypodion species. This composition shows similarities to some mammal microbiomes (Ley et al., 2006; Gomez et al., 2017). The phylum Firmicutes (synonymous with Bacillota) are a classification of gram-positive bacteria deriving their name from their thick cell walls (Gibbonst & Murray, 1978). Studies on mice have identified Firmicutes as an integral component of the digestive tract, playing a large role in the release of calories from food items (Ley et al., 2006). Their dominance in the hindguts of chameleon may allude to a beneficial role in the digestive process. Beyond this beneficial relationship, the bacterial phylum may be ideally suited to the harsh conditions of the mid- to hind-gastrointestinal tract as several genera (e.g., Bacillus and Clostridium) within the Firmicutes phylum can form endospores to survive adverse environmental conditions (Egan et al., 2021). This could explain why Firmicutes are ubiquitous in many vertebrate digestive tracts.

The bacterial phylum Bacteroidota also displayed a high frequency within samples across all three species. Bacteroidota (synonymous with Bacteroidetes) is a classification of gram-negative bacteria including a diverse array of species (Rajilić-Stojanović & de Vos, 2014). The phylum is exceptionally abundant and well described in the gastro-intestinal tract; performing a myriad of beneficial functions (Marcobal et al., 2011; Rajilić-Stojanović & de Vos, 2014). This would explain the high frequency of bacteria belonging to Bacteroidota in dwarf chameleon digestive tracts.

The consistent prevalence of Verrucomicrobiota in the faecal samples of dwarf chameleons represents a unique opportunity for future study of this poorly described bacterial group. Verrucomicrobiota was formally described in 1997 as a gram-negative bacterial phylum that contains the amino acid diaminopimelic acid, with members being highly fimbriated and sometimes producing prosthecae (Hedlund, Gosink & Staley, 1997). This phylum has been noted to be common in marine (Orellana et al., 2022) and soil (Bergmann et al., 2011) environments, as well as in the intestinal mucosa of human digestive tracts (Dubourg et al., 2013; Fujio-Vejar et al., 2017). Despite its ubiquity this bacterial phylum is still known to a relatively poor degree. In soil environments some members of Verrucomicrobiota act as saprotrophs for cellulose rich organic matter (Rakitin et al., 2024). The role of Verrucomicrobiota in digestive tracts still is not entirely understood, however in termites it may play a role in cellulose, starch, and sugar digestion (Wertz et al., 2012).

The identification of the Rs-K70 termite group candidate phylum in Bradypodion digestive tracks gives the potential for microbiome studies to further elucidate dietary relationships in organisms. This candidate phylum is known to occur in termites and dendrophagus cockroaches (both in the order Blattodea) (Herlemann, Geissinger & Brune, 2007; Diouf et al., 2015), which likely suggests that these insects form a notable part of the diets of dwarf chameleons. Previous research on dwarf chameleon diets have identified Blattodea within the stomach contents (Measey et al., 2011; Carne & Measey, 2013; da Silva et al., 2016) suggesting a link between the prey item and the predator microbiome. This link between food items and microbiome composition could be used as a monitoring tool for prey availability in an immediate area. This could help to conserve populations by acting as an indication of whether the ecosystem has the ability to continue supporting the population or whether it is worthwhile implementing conservation measures (such as translocations).

Focused examination of the identified ASVs found representative identifications associated to Campylobacter, Escherichia, and Serratia. The frequencies of these genera, however, were low and only one could be confidently identified beyond genus level. Furthermore, these bacterial genera are not unique to reptiles and are ubiquitous in many organisms, often playing beneficial roles. For example, one ASV was identified as Serratia symbiotica which is a bacterium that is a beneficial symbiont found in aphids (Lamelas et al., 2011; Moran et al., 2005) suggesting a likely source from the diet of chameleons. Searches of Campylobacter found one ASV identification that shared the same order (Campylobacterales) which was identified as an Arcobacter. Arcobacter bacteria have been recognised as an emerging food borne zoonotic pathogen (Ramees et al., 2017). This relationship, however, is species dependent and requires consumption of an infected food source making their presence in chameleons of limited concern. Furthermore, their extremely low prevalence only in the faecal dataset likely indicates Arcobacter is likely not commonplace in chameleons and this singular identification likely arises from an environmental or food source. Searches of Escherichia found four ASV identifications assigned to the ‘Escherichia-Shigella’ genus complex. Although some bacterial species belonging to this complex are known to be pathogenic (causing diarrhoea in humans Phiri et al., 2021) and zoonotic (Ramatla et al., 2022) the overwhelming majority of species are harmless and are typically required for a healthy digestive tract in humans as well as other animals (Martinson & Walk, 2020). Overall, zoonotic microorganisms associated with the ASVs identified in chameleons are extremely low in prevalence, and with no notable links to pathogenicity. This implies that zoonotic transfer from chameleons is nominal at worst.

Habitat effects on microbiome composition

Comparison of the microbiome’s compositions between urban and natural populations of the three Bradypodion species suggest that host-species of origin is a pronounced driver of microbial community composition whilst urbanisation has a much smaller overall effect. This substantial difference may arise from the fact that urbanisation has existed for a relatively short period of time (last few hundred years) compared to the time of species divergence (ca. 1.5–11 mya; Tolley, Chase & Forest, 2008). Despite the shorter period that urbanisation has had to influence the microbiome, small distinctions were recognised between the populations to varying degrees.

The dominant bacterial phyla (i.e., Firmicutes, Proteobacteria, and Bacteroidota) were conserved between populations, but overall frequency, and level of variability between populations were unique to each species. This may be a result of functional shifting in the microbiome assemblages, allowing bacteria within Firmicutes, Proteobacteria, and Bacteroidota to partially overtake each other’s physiological functions when needed. This would be highly dependent on the specific roles that the groups are playing in the digestive tract. There is, however, already potential for partial overlap based on the similar metabolic and digestive roles these three phyla are known to participate in during digestion (Marcobal et al., 2011; Rajilić-Stojanović & de Vos, 2014; Moon et al., 2018). The potential health impacts of these shifts are not understood, however, changes in natural habitats may expose dwarf chameleons to dysbiosis (Schumacher, 2006).

Significance testing of microbial group dispersion (PERMDISP) indicated that microbial assemblages are mirrored between urban and natural populations, suggesting habitat changes are not causing shifts in the microbiomes. This could indicate that vertical transmission (e.g., Baca et al., 2012) of microbiota is more influential than the environment in dictating chameleon microbiomes. Alternatively, environmental microbiota may be highly homogeneous between the habitats; however, an independent examination would need to occur to determine whether this is the case. Microbial group dispersion was indicated to be different in the hindguts of B. thamnobates individuals both in terms of core taxa (i.e., Firmicutes, Proteobacteria, and Bacteroidota) and low frequency taxa. A likely cause of this difference would be alteration of dietary items, which can cause shifts in gut microbiota (Hicks et al., 2018; Dion-Phénix et al., 2021). Furthermore, as urbanisation and habitat transformation have been shown to have complex impacts on arthropod communities, changing both the overall arthropod diversity (Adams et al., 2020; Theodorou et al., 2020) as well as predator–prey interactions (Rocha & Fellowes, 2018), it is reasonable to assume that this is a likely driver of differences in gut microbiota of their predators (e.g., chameleons).

Significance in microbial community variation (PERMANOVA) was noted for all tested beta diversity metrics that examined faecal samples from B. melanocephalum, whilst B. setaroi and B. thamnobates showed minimal variations. The higher-level of microbiome variation between populations from different habitats in B. melanocephalum may be due to the clear distinction between the urban (roadside and garden trees) and natural (grassy savanna) vegetations (Fig. 1 & Fig. S1). The lack of differences for B. thamnobates and B. setaroi could be explained by similarities between urban and natural habitat vegetation, which would limit potential microbial assemblage shifts. Indeed, although the habitats for these two species are categorised as urban, the habitat (vegetation) differentiation between natural and urban sampling sites for B. setaroi was minimal, as much of the urban town seamlessly blends into the natural forests surrounding it (Fig. S2) with ample forest patches still within the town. For B. thamnobates, the natural habitat sampled was typically good-quality, but consisted of spatially separated forest patches, within a larger matrix of transformed agricultural habitat and small urban settlement (Fig. S3), with chameleons sampled from alongside road-verges or private gardens. Overall, the strongest microbiome differences occurred between populations with the highest habitat disparity between urban and natural habitats.

Future possibilities and current limitations

This study set out with the goals of creating the initial microbiome description for three species within Bradypodion as well as examining how this microbiome is affected in light of the devastation of anthropogenic modification. Although these outcomes were achieved the potential for future research on dwarf chameleon microbiomes is great. Some of this potential research has already been indicated in light of microbiome overlaps between Anolis and Bradypodion, using Bradypodion as a case study group for the Verrucomicrobiota phylum, and examining whether the current dataset is constrained in the identification of zoonotic bacteria. Furthermore, to more strongly determine the similarity or disparity between microbiome composition and habitat type repeated sampling between populations located at differing distances could be conducted to ensure microbiome homogenisation is not a potential confounding factor. There are, however, several other areas that could be explored.

For example, the examination of microbial assemblages in the context of their host phylogeny has given rise to the concept of phylosymbiosis (Lim & Bordenstein, 2020; Mallott & Amato, 2021). This phylogenetic symbiosis occurs when a host’s phylogeny is reintegrated in the microbiome of the species that make up its lineages. The present study made use of three of the 20 Bradypodion species (Tolley, Burger & Tilbury, 2022) and insights into the potential of phylosymbiosis within the genus were limited. A more detailed exploration of the genus would be ideal at uncovering the potential for phylosymbiosis.

A further application of the study of the microbiome in dwarf chameleons could target the behavioural and social implications for shifts in the microbiota. The link between microbiomes and behaviour is still an emerging field with many aspects yet to be fully explored; however, changes in microbiota assemblages have strong ties to neurological development and behaviour (overview in Vuong et al., 2017). The potential impact this could have on chameleons is vast and could inform both conservation of wild populations, as well as handling of captive populations. Further studies would benefit from examining behaviour, behaviour linked to microbiome assemblage, dysbiosis, and neurological impacts of microbiome shifts in dwarf chameleons.

Lastly, both the presence and success of Bradypodion within urban landscapes could make the genus a potential bioindicator of habitat health and diversity in transformed areas. This concept has been applied to other organisms in light of their environmental sensitivity (e.g., mosses used to gauge air pollution: Radziemska et al., 2019). A possible approach builds on the functional links between diet and chameleon microbiome displayed above. This could use fluctuations in chameleon microbiome composition to gauge ecosystem diversity in an immediate area with a less invasive sampling than traditional insect capturing. This would essentially make use of the natural hunting behaviour of the organism as an ecosystem sampler (in this case mainly of arthropod diversity).

Conclusions

The present study generated the first descriptions of the microbiomes within dwarf chameleon foreguts and hindguts. Our results demonstrated that bacterial diversity of dwarf chameleon microbiomes showed high similarity to other squamates, indicating that Bradypodion microbiomes are typical for this order of reptiles. Exploration of zoonosis in dwarf chameleons found essentially no potential within the genus, implying that zoonotic transfer from chameleons is not expected. Future work would also benefit from more expansive sampling targeting other species within the genus, as well as examination of viral, fungal, archaeal, and protozoic microbiome diversity. Furthermore, the present study examined the effect of urbanisation on the microbiomes of dwarf chameleons. Our results demonstrated that the microbiome assemblages of B. melanocephalum were more divergent between populations compared to B. setaroi or B. thamnobates. The hypothesised reason for this discrepancy is the variation in vegetation type occurring in each habitat. This trend between natural and urban populations of dwarf chameleons led to the conclusion that microbial community variations are associated with distinctness of habitat. Future research would also benefit from examining the presence of bacterial taxa (such as Rs-K70 termite group and Serratia symbiotica) to dietary links in prey items (e.g., arthropods) as a potential source for microbiome homogenisation.

Supplemental Information

Supplemental Information 1 Sample localities for Bradypodion melanocephalum.

Localities are coloured yellow from urban habitat and light blue from natural habitat. Faecal samples are indicated by crosses and buccal swab samples are indicated by triangles.

Supplemental Information 2 Sample localities for Bradypodion setaroi.

Localities are coloured yellow from urban habitat and light blue from natural habitat. Faecal samples are indicated by crosses and buccal swab samples are indicated by triangles.

Supplemental Information 3 Sample localities for Bradypodion thamnobates.

Localities are coloured yellow from urban habitat and light blue from natural habitat. Faecal samples are indicated by crosses and buccal swab samples are indicated by triangles.

Supplemental Information 4 Beta diversity plots comparing habitat occupation in Bradypodion melanocephalum.

Principal Coordinate Analysis (PCoA) for calculated beta diversity metrics (Bray-Curtis dissimilarity index, Jaccard similarity index, Unweighted UniFrac distance, and Weighted UniFrac distance) across buccal swab samples (left) and faecal material samples (right) characterised by natural and urban populations from Bradypodion melanocephalum.

Supplemental Information 5 Beta diversity plots comparing habitat occupation in Bradypodion setaroi.

Principal Coordinate Analysis (PCoA) for calculated beta diversity metrics (Bray-Curtis dissimilarity index, Jaccard similarity index, Unweighted UniFrac distance, and Weighted UniFrac distance) across buccal swab samples (left) and faecal material samples (right) characterised by natural and urban populations from Bradypodion setaroi.

Supplemental Information 6 Beta diversity plots comparing habitat occupation in Bradypodion thamnobates.

Principal Coordinate Analysis (PCoA) for calculated beta diversity metrics (Bray-Curtis dissimilarity index, Jaccard similarity index, Unweighted UniFrac distance, and Weighted UniFrac distance) across buccal swab samples (left) and faecal material samples (right) characterised by natural and urban populations from Bradypodion thamnobates.

Supplemental Information 7 List of 1,065 retained ASV identifications across all buccal swab samples for Bradypodion

Shown are taxonomic rank, taxonomic nomenclature of identification, the confidence of identification, and the unique sequence ASV identifier assigned. Any uncultured or unidentified classifications were assigned to the next highest taxonomic rank that was certain.

Supplemental Information 8 List of 4,458 retained ASV identifications across all faecal material samples for Bradypodion

Shown are taxonomic rank, taxonomic nomenclature of identification, the confidence of identification, and the unique sequence ASV identifier assigned. Any uncultured or unidentified classifications were assigned to the next highest taxonomic rank that was certain.

Thanks go to the South African National Biodiversity Institute (SANBI), and Shilpa Parbhu for administrative and logistical support. We would also like to thank Azraa Ebrahim, Juan-Jacque Forgus, Devon Main, Melissa Petford, Kirsten Stephens, and Jody Taft for assistance in the field, as well as Graham Alexander for project support, Tristan Peebles for illustration support, and Gerrit Botha for bioinformatic support. We acknowledge the use of the ilifu cloud computing facility (http://www.ilifu.ac.za), a partnership between the University of Cape Town, the University of the Western Cape, Stellenbosch University, Sol Plaatje University, the Cape Peninsula University of Technology, and the South African Radio Astronomy Observatory. The ilifu facility is supported by contributions from the Inter-University Institute for Data Intensive Astronomy (IDIA, a partnership between the University of Cape Town, the University of Pretoria, and the University of the Western Cape), the Computational Biology division at UCT and the Data Intensive Research Initiative of South Africa (DIRISA).

Additional Information and Declarations

Competing Interests

Author Contributions

Animal Ethics

DNA Deposition

Data Availability

The authors declare there are no competing interests.

Matthew G. Adair conceived and designed the experiments, performed the experiments, analyzed the data, prepared figures and/or tables, authored or reviewed drafts of the article, and approved the final draft.

Krystal A. Tolley conceived and designed the experiments, authored or reviewed drafts of the article, and approved the final draft.

Bettine Jansen van Vuuren conceived and designed the experiments, authored or reviewed drafts of the article, and approved the final draft.

Jessica Marie da Silva conceived and designed the experiments, authored or reviewed drafts of the article, and approved the final draft.

The following information was supplied relating to ethical approvals (i.e., approving body and any reference numbers):

All animal handling and sample collection was approved by the University of Johannesburg (ethics no.: 2019-10-10/van Vuuren_Tolley) and was carried out under research permits from Ezemvelo KZN Wildlife (OP2635/2020) and a research agreement with the iSimangaliso Wetland Park Authority.

The following information was supplied regarding the deposition of DNA sequences:

New DNA sequences generated for this study are available at GenBank: PRJNA1184952.

The sequence data is also available on FigShare: Adair, Matthew; Tolley, Krystal; Jansen van Vuuren, Bettine; da Silva, Jessica (2024). Bradypodion Microbiome Raw Sequence Data. figshare. Dataset. https://doi.org/10.6084/m9.figshare.26096539.v1.

The following information was supplied regarding data availability:

The amplicon sequence variant identifications made during analysis and the sample maps are available in the Supplementary Files.

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
