# Peer review of "Anthropogenic reverberations on the gut microbiome of dwarf chameleons (Bradypodion)"

_PeerJ, doi:10.7717/peerj.18811_

## Round 0.1 · original submission · Major Revisions

Please address issues pointed by all reviewers and revise your manuscript accordingly.

·

Basic reporting

Staff Editor
PeerJ Life & Environment
I am writing to express my gratitude for considering me to review the scientific manuscript titled "The microbial reverberations of living in human-made
environments on the digestive-tract microbiomes of dwarf chameleons (Bradypodion)" for publication in PeerJ. It is an honor to contribute to the review process. In this MS the relationship between habitat occupation and microbiome composition was explored in
three Bradypodion species to assess whether occupation of transformed habitats has influenced the bacterial microbiome composition in the dwarf chameleon digestive tract.
The following is the revision of the MS.
1. Basic reporting.
The manuscript is written in clear English, making it easy to follow the authors' arguments and findings. The authors have provided a comprehensive review of the relevant literature, citing key studies that offer a sufficient context for the research. The manuscript structure adheres to PeerJ standards, including a well-organized abstract, introduction, methods, results, discussion, and conclusion sections. Some figures and tables are well-designed and clearly presented. However, the colours chosen for the bar charts in Figures 2 and 3 do not allow for differentiation between the different phyla identified in the samples. Please select another colour scheme for these figures.
The legends are detailed and provide necessary explanations. ASVs data has been shared as supplementary material, which enhances the transparency and reproducibility of the study.

Experimental design

The manuscript presents an original research study that aligns well with the scope of the journal. The research question “are there differences in the microbial diversity between natural and urban
populations of dwarf chameleons from KwaZulu-Natal?” is clearly defined, addressing a significant gap in the existing knowledge. The investigation is conducted with rigor, adhering to high technical and ethical standards. The methods are described in sufficient detail that allows for replication of the study. Overall, the experimental design is robust and well-executed.
In regards to sample collection, I wonder, given the short distances between what the authors defined as natural and urban environments, if it is possible for chameleons to move from the city to the countryside or vice versa?

Validity of the findings

While the authors present valuable 16S rRNA gene sequencing data and have conducted some appropriate data analyses, a critical limitation of the study is the absence of statistical testing to evaluate the significance of observed differences in bacterial community composition. For instance, although variations in bacterial phylum frequency between urban and natural populations are reported (Table 1), the lack of statistical support hinders definitive conclusions about the importance of these differences. To address this, I recommended conducting robust statistical analyses on the relative abundance of bacteria at multiple taxonomic levels (phylum, class, order, etc.). Such analyses will provide a solid foundation for comparing and drawing conclusions about the microbial communities associated with rural and natural chameleon environments. These statistical findings should be integrated into the discussion and conclusions to support the acceptance or rejection of the study's hypotheses."

Minor revision:
Material and methods:
- Please specify the amount of faecal samples used for DNA extraction.
Results:
- Please write the numbers in the correct format, without spaces separating millions from thousands and thousands from hundreds. Ex: replace 10 358 081 (line 232 page 12) with 10.358.081 or 10358081.
- Lines 236-237: please delete in these lines the sentence “515f-806r-animal-distal-gut-classifier.qza (Quast et al., 2013, Kaehler et al., 2019, Robeson et al., 2021))”. It was already included previously in the M&M section.

- Lines 240-252: Please include the percentage abundance of each Phylum within parentheses to provide more detailed results.

- It is unclear why Figures 4, 5, and 6 (beta diversity plots) were included in the manuscript as their corresponding results are not discussed in the results section. One more time, regarding beta diversity indices, it is very important to performed statistical analysis in order to determine if the beta diversity plots correlate with Table 2 (PERMANOVA for urban and natural populations of each Bradypodion species).

- To fully understand the nature of the observed differences in urban and natural populations for faecal samples of B. melanocephalum (Table 2), a more in-depth analysis is warranted. By performing a differential abundance analysis using an appropriate tool like DESeq2, we can identify the specific bacterial taxa that contribute to these differences

Discussion and conclusion
The discussion section of the manuscript should be revised in light of the new results obtained from the requested statistical analyses.

Additional comments

Given the absence of robust statistical analyses, the need for more in-depth analysis of the raw data, and other identified shortcomings, the manuscript requires major revisions before it can be considered for publication in PeerJ. The lack of adequate statistical analyses hinders a solid evaluation of the results and limits the ability to generalize conclusions.

Reviewer 2 ·

Basic reporting

This study is crucial for reptile research as it establishes baseline microbiome data for dwarf chameleons, providing essential insights and reference points for future studies. It meets the journal's standards and significantly contributes to the field

Experimental design

The experimental design was sufficient and complied with the journal's standards

Validity of the findings

No comment

Additional comments

Please find my comment in the manuscript

Annotated reviews are not available for download in order to protect the identity of reviewers who chose to remain anonymous.

Reviewer 3 ·

Basic reporting

Title
The title of the given manuscript should be revised and made compact, attention-capturing, and interesting to the reader
Abstract:
The abstract should highlight the overall contribution of the study at the end in a clear way
Introduction
The introduction should be arranged in such a way that the last paragraph should be a concluding one. The practical application of the study should be described clearly.
Discussion:
• In the discussion opposing references should be focused.
• At the end of the discussion, the future research directions for new researchers should be mentioned.
• check for the quality of English and use simple and plain language.
• Try to use more specific and clear wording, don’t use ambiguous type sentences
• There is some lack of in-depth discussion
Conclusion
It should be conclusions. The conclusion does not reflect the overall manuscript. It should be rewritten.
Clarity and Structure:
• Some lack of clarity in the results and discussion section
• There are some typos or grammatical errors
Recommendations:
Need major revision, should be focused on English language, quality of words, clarity, and all other above-mentioned suggestions.

Experimental design

Methodology:
• The sample transportation details are missing
• A map or location diagram can be added to tell the specific area of your study.
Experimental design
Can be presented in a pictorial or tabulated form to make it clearer

Validity of the findings

no comments

Additional comments

no additional comments

Annotated reviews are not available for download in order to protect the identity of reviewers who chose to remain anonymous.

---

## Round 0.2 · Minor Revisions

Please address remaining concerns of the reviewer #2 and make the necessary amendments in the manuscript.

Reviewer 2 ·

Basic reporting

no comment

Experimental design

no comment

Validity of the findings

no comment

Additional comments

The study provides essential baseline microbiome data for dwarf chameleons, meeting journal standards and offering significant contributions to reptile research. The experimental design is adequate and aligns with the journal's requirements. The authors have made the amendments accordingly.

Reviewer 3 ·

Basic reporting

• Language and Clarity: The manuscript is generally written in clear and professional English. However, minor grammatical corrections and rephrasing could enhance clarity in certain sections, such as the methodology and conclusions. For instance, the phrase “The buccal cavity microbiomes between the three species of Bradypodion were remarkably homogeneous” could be rewritten for smoother readability.
• Literature Context: The references cited are comprehensive and relevant, offering sufficient background for the study's focus. The manuscript effectively situates the research within the broader context of microbiome studies and reptilian conservation.
• Structure: The article follows a standard scientific structure. Figures and tables are high-quality, appropriately labeled, and provide relevant information. Supplementary materials and raw data files are appropriately linked and referenced.

Experimental design

• Scope and Originality: The research falls within the journal's scope, addressing an unexplored topic—the gut microbiome of dwarf chameleons under urban and natural habitat conditions. The study identifies a clear knowledge gap, as this microbiome has not been previously characterized.
• Methodological Rigor: The methodology is rigorous, employing well-established techniques like 16S rRNA sequencing for microbiome analysis. Ethical approvals and field permits are explicitly mentioned, ensuring compliance with ethical standards.
• Detail and Replicability: Most methods are described in sufficient detail for replication. However, it would be helpful to include more specifics about the statistical software and parameters used in PERMANOVA tests and ANCOM analyses to ensure replicability.

Validity of the findings

• Data and Statistical Robustness:The data presented is robust, and statistical analyses are well-applied to support the findings. The use of alpha and beta diversity metrics, along with ANCOM, provides a comprehensive understanding of microbial differences. The manuscript acknowledges limitations in sampling depth and the need for further studies to confirm the trends observed.
• Conclusions: The conclusions are well-supported by the data, linking microbial community variations to habitat distinctions and species-specific traits. However, some claims, such as potential zoonotic implications, could benefit from further elaboration or tempering to avoid overinterpretation.

Additional comments

• Strengths:The study provides foundational data on the microbiomes of Bradypodion species, with implications for conservation biology and habitat management.The manuscript’s integration of urban ecology with microbial diversity is timely and significant.
• Weaknesses: More in-depth discussion on the implications of microbiome differences between urban and natural habitats is needed, particularly regarding conservation strategies. Minor inconsistencies in formatting and phrasing (e.g., “16 Figure file(s)” mentioned in reviewer guidance but referenced inconsistently).
• Suggestions for Improvement:Expand the discussion of Verrucomicrobiota and its ecological roles. Provide a clearer connection between findings and potential conservation applications, especially regarding Bradypodion as bioindicators.

---

## Round 0.3 · accepted · Accept

All remaining concerns were adequately addressed and revised manuscript is acceptable now.